# Identification of the Hepatic Metabolites of Flumazenil and their Kinetic Application in Neuroimaging

**DOI:** 10.3390/ph16050764

**Published:** 2023-05-18

**Authors:** Wei-Hsi Chen, Chuang-Hsin Chiu, Shiou-Shiow Farn, Kai-Hung Cheng, Yuan-Ruei Huang, Shih-Ying Lee, Yao-Ching Fang, Yu-Hua Lin, Kang-Wei Chang

**Affiliations:** 1Isotope Application Division, Institute of Nuclear Energy Research, Taoyuan City 325207, Taiwan; whchen@iner.gov.tw (W.-H.C.); amanda@iner.gov.tw (S.-S.F.);; 2Department of Nuclear Medicine, Tri-Service General Hospital, Taipei 114202, Taiwan; treasure316@gmail.com; 3Taipei Neuroscience Institute, Taipei Medical University, Taipei 110301, Taiwan; eugene_8888@yahoo.com.tw; 4Laboratory Animal Center, Taipei Medical University, Taipei 110301, Taiwan; yvette8816@tmu.edu.tw

**Keywords:** flumazenil, LC-QqQ MS, metabolism, γ-aminobutyric acid receptor antagonist, benzodiazepine receptors

## Abstract

Studies of the neurobiological causes of anxiety disorders have suggested that the γ-aminobutyric acid (GABA) system increases synaptic concentrations and enhances the affinity of GABA_A_ (type A) receptors for benzodiazepine ligands. Flumazenil antagonizes the benzodiazepine-binding site of the GABA/benzodiazepine receptor (BZR) complex in the central nervous system (CNS). The investigation of flumazenil metabolites using liquid chromatography (LC)-tandem mass spectrometry will provide a complete understanding of the in vivo metabolism of flumazenil and accelerate radiopharmaceutical inspection and registration. The main goal of this study was to investigate the use of reversed-phase high performance liquid chromatography (PR-HPLC), coupled with electrospray ionization triple-quadrupole tandem mass spectrometry (ESI-QqQ MS), to identify flumazenil and its metabolites in the hepatic matrix. Carrier-free nucleophilic fluorination with an automatic synthesizer for [^18^F]flumazenil, combined with nano-positron emission tomography (NanoPET)/computed tomography (CT) imaging, was used to predict the biodistribution in normal rats. The study showed that 50% of the flumazenil was biotransformed by the rat liver homogenate in 60 min, whereas one metabolite (M1) was a methyl transesterification product of flumazenil. In the rat liver microsomal system, two metabolites were identified (M2 and M3), as their carboxylic acid and hydroxylated ethyl ester forms between 10 and 120 min, respectively. A total of 10–30 min post-injection of [^18^F]flumazenil showed an immediate decreased in the distribution ratio observed in the plasma. Nevertheless, a higher ratio of the complete [^18^F]flumazenil compound could be used for subsequent animal studies. [^18^F] According to in vivo nanoPET/CT imaging and ex vivo biodistribution assays, flumazenil also showed significant effects on GABA_A_ receptor availability in the amygdala, prefrontal cortex, cortex, and hippocampus in the rat brain, indicating the formation of metabolites. We reported the completion of the biotransformation of flumazenil by the hepatic system, as well as [^18^F]flumazenil’s potential as an ideal ligand and PET agent for the determination of the GABA_A_/BZR complex for multiplex neurological syndromes at the clinical stage.

## 1. Introduction

Previous preclinical studies suggested that increased synaptic γ-aminobutyric acid (GABA) concentrations enhance the affinity of GABA_A_ receptors for benzodiazepine ligands [1,2]. Flumazenil antagonizes the benzodiazepine binding site of the GABA/benzodiazepine receptor complex (BZR) in the central nervous system (CNS), thereby preventing chloride channel opening and inhibiting neuronal hyperpolarization [3,4]. The imidazo-benzodiazepine derivative flumazenil, or ethyl 8-fluoro-5-methyl-6-oxo-5,6-dihydro-4H-benzo[f]imidazo [1,5-a] [1,4]diazepine-3-carboxylate (molecular formula: C_15_H_14_FN_3_O_3_; molecular weight: 303), was initially launched in 1987 by Roche under the trade name Anexate^®^. Subsequently, it was confirmed to be an antagonist specific to benzene diazonium salts. Its mechanism works by competing with the benzene diazonium salt for the GABA_A_ receptors [3,5,6,7]. Intravenous flumazenil treatment is used in the recovery from anesthesia, reducing the hypnotic and sedative effects of benzodiazepines via competitive inhibition at the benzodiazepine-binding site on the GABA_A_ receptor. In the field of nuclear medicine, isotope-labeled flumazenil could become a useful radiopharmaceutical for the early diagnosis of anxiety disorders in the central nervous system.

In animals, foreign chemicals are metabolized mainly in the liver [8]. Hepatic enzyme metabolism studies have been conducted using various hepatic components including supersomes, microsomes, cytosolic fractions, cell lines, hepatocytes, and liver slices [9]. Despite being entirely composed of cytochrome P450 (CYP) and uridine glucuronosyl transferase (UGT) enzymes, the liver microsome is the most popular model because it is easy to apply, can be commercialized, is affordable, and is well established. This model provides an incomplete representation of the in vivo situation and renders the results unsuitable for quantitative evaluation. Most drug metabolites reduce biological activity, degrade the molecules into smaller molecules, and increase polarity and water solubility to facilitate excretion in the urine [9]. Lavén et al. studied the metabolic stability of flumazenil using a capillary liquid chromatography-mass spectrometry-based method [10]. Flumazenil is mostly metabolized in the liver, and metabolites from the liver microsomal enzyme catalysis of ^11^C-flumazenil and its ^18^F-analogue (fluoroethyl-flumazenil) were identified using ultra-high performance liquid chromatography (UHPLC)/quadrupole-time-of-flight-mass spectrometry (Q-ToF-MS) and high performance liquid chromatography (HPLC)/(ESI-QqQ MS), respectively [11,12].

An increase in the binding capacity of the GABA_A_ benzodiazepine-receptor site-specific radioligand (^18^F-flumazenil), using molecular imaging tracers to visualize the distribution of GABA_A_ receptors in the brain, has been employed to aid in the diagnosis of psychiatric/CNS disorders (such as epilepsy, anxiety, and insomnia) [6,13]. It can be traced by radioactive flumazenil derivatives that are more sensitive and accurate than 2-deoxy-2[^18^F]fluoro-D-glucose ([^18^F]F-FDG) imaging that is used for epileptic foci localization [5,7]. In the present study, a convenient non-carrier labeling F-18 with flumazenil and a radio-HPLC unit were used for synthesis. [^18^F]flumazenil was used for in vivo, ex vivo, and in vitro analysis. The investigation of flumazenil metabolites into rat liver homogenates and microsomes was analyzed using liquid chromatography-tandem mass spectrometry. This study aimed to clarify the biotransformation of flumazenil using in vivo and in vitro hepatic systems and to quantify the GABA_A_/BZR complex distribution of [^18^F]flumazenil by nano-positron emission tomography (NanoPET). When fully characterized, PET-based GABA_A_/BZR images provide potential biomarkers for the diagnosis and treatment of patients with multiple neurological syndromes.

## 2. Results

### 2.1. High Performance Liquid Chromatography Analysis of Flumazenil

The ZORBAX Eclipse XDB-C18 column (100 × 4.6 mm, 5 μm) separated biotransformed flumazenil from the hydrophilic matrices of the liver microsome and the liver homogenate solutions. The mobile phase was composed of aqueous ammonium acetate buffer and acetonitrile (ACN) programmed gradient eluent (Table 1). After pretreatment, the typical flumazenil chromatogram exhibited a spike in the rat liver homogenate (Figure 1). The retention time (R_t_) for flumazenil was 8.2 ± 0.1 min. The column efficiency exceeded 8000 p/m, with a tailing factor of 1.1 (calculated using Agilent ChemStation software 10.02), and the flumazenil peak was far from the hepatic matrix peaks, with good resolution to avoid contamination. The dynamic range of flumazenil in the hepatic matrices was 1–100 ppm, with a linear least-squares regression equation of *Y* = 35.4*X*—18. The calibration curve (six concentration points without being forced through zero) for flumazenil was sufficiently linear, with a correlation coefficient (r^2^) of 0.9992. The limits of detection (LOD) and quantification (LOQ) for flumazenil were 0.1 and 0.4 ppm, respectively, using DAD at λ of 250 nm. The incubated solutions were analyzed using HPLC to separate flumazenil and its metabolites. The peak area of flumazenil was measured in triplicate and then averaged to investigate and determine the reaction trend of flumazenil in hepatic biosystems.

### 2.2. Mass Spectrometric Analysis of Flumazenil

In the ESI spectrum of flumazenil, the [M + H]^+^ and [M + Na]^+^ ions were observed at *m*/*z* 304 and 326, respectively. Product-ion scans were performed under different collision-activated dissociation conditions to optimize the declustering potential, entrance potential, collision energy, and collision-cell exit potential. A total of 304 product ions were observed at *m/z* values of 276, 258, 229, 217, and 189 (Figure 2A). Fragment structures were used to clarify the fragmentation of flumazenil and assist with metabolite identification (Figure 2B). The most abundant product ions were 258 and 276, indicating metastable de-ethoxylation (Δ*m*/*z* = −46) and de-ethylation (Δ*m*/*z* = −28) from the parent molecular ion 304 to form acylium and protonated carboxylic acid, respectively.

### 2.3. Study of Flumazenil Biotransformation by Hepatic Enzyme Systems

#### 2.3.1. Flumazenil Metabolism Study in Rat Liver Homogenate

After incubation in the rat liver homogenate for 240 min, the broths were pretreated to separate the flumazenil concentrate and its derivatives from the hepatic matrices. This was followed by an HPLC analysis of the overlapping chromatograms of flumazenil before and after incubation in the liver homogenate for various durations. The peak height of the parent drug was reduced, while a metabolite peak 1 (M1) appeared with (R_t_ = 7.45 ± 0.1 min) and gradually increased in the peak area (Figure 3A). The flumazenil biotransformation rate in the rat liver homogenate was evaluated based on the trends in the peak areas of flumazenil in the chromatograms (Figure 3B). These trends showed that the rate of flumazenil biotransformation in the rat liver homogenate was rapid during the first 30 min, but subsequently slowed down to approximately 50% at 60 min. After 4 h, approximately 65% of the parent drug was biotransformed and the *m*/*z* of M1 was 290 amu less 14 (lost CH_2_) than that of flumazenil, whereas the core structures of M1 and flumazenil were identical (Figure 3C). Therefore, M1 was the methyl ester of flumazenil. The metabolic pathway of flumazenil in rat liver homogenates is thought to be methyl transesterification.

#### 2.3.2. Flumazenil Metabolism Study in Human Liver Microsomes

The biotransformation of flumazenil has also been studied in rat hepatic microsomes. The 2 h reactions were conducted according to the directions accompanying the supplied microsomes. The established LC-DAD method was used to evaluate the flumazenil in the reaction solutions. The time course of flumazenil and its metabolites in rat hepatic microsomes was plotted based on the average chromatographic peak areas identified from experiments conducted in triplicate. Two metabolite peaks, 2 and 3 (M2, major, and M3, minor), were identified in the rat liver microsomal system, with Rt values of 6.5 and 7.7 min, respectively (Figure 4A,B). Flumazenil was detected at approximately 22% of the initial amount of rat liver microsomes, with up to a 120 min incubation time, and the peak area of M2 increased by 72%. No significant increase in the peak area of M3 was observed (Figure 4C).

### 2.4. A Summary of the Metabolic Pathways of Flumazenil

The metabolic pathways of flumazenil in rat liver homogenates and the microsomal enzymatic system are summarized in Figure 5. In this study, the flumazenil metabolic pathways were identified in rat and human liver microsomes. Differences in metabolic pathways between liver microsomes and whole liver homogenates arise from variations in enzyme expression levels, efficiencies, and reactants in the media. Biotransformation of the flumazenil ester into the carboxylic acid form generally occurs in another hepatoma radiopharmaceutical, ^188^Re-MN16ET. In addition to the identification of the flumazenil metabolites M2 and M3, we also detected another metabolite, M1, in the rat liver media; this appeared to be a methyl transesterification product of flumazenil, which has not been discussed previously in the literature. This methyl replacement of the ethyl ester is thought to be an undisclosed flumazenil metabolite.

### 2.5. In Vitro Serum Stability of [^18^F]flumazenil

This procedure resulted in a higher quality [^18^F]flumazenil with a radiochemical purity > 95% after radio-HPLC and radio-thin layer chromatography (TLC). In an amber glass vial, the radiochemical purity of [^18^F]flumazenil in 30% ethanol was >90% at 8 h after the end of synthesis (EOS) at room temperature. A total of 3.7 MBq of [^18^F]flumazenil was incubated in Sprague Dawley (SD) rat exsanguinated serum. The in vitro serum stability of the radiochemical purity (RCP%) values of [^18^F]flumazenil were 95.76%, 93.28%, 91.07%, 87.46%, 85.68%, 79.95%, and 72.13% at 0, 5, 10, 30, 60, 120, and 240 min, respectively. Our in vitro serum stability results showed that intact [^18^F]flumazenil was >85% at up to 1 h of incubation, under regular conditions (Table 2).

### 2.6. In Vivo NanoPET/CT Assay of [^18^F]flumazenil

In the [^18^F]flumazenil/NanoPET/CT images, radioactivity was detected with the GABA_A_ receptor in all brain regions of the rats (Figure 6A). The region of interest (ROIs) were extracted from a set of previously constructed regions, including the frontal cortex, striatum, hippocampus, amygdala, midbrain, cerebellum, and pons (Figure 6B). This phenomenon was observed in the 30–40 min post-injection timeframe, which displayed significant differences in specific binding ratios (SBRs = target region-reference region)/reference regions) in the frontal cortex, cortex, amygdala, and hippocampus. At this distribution time, the SBRs of these target regions were 2.07~3.32 times higher than the radiation accumulated in the medulla (reference region).

### 2.7. Ex Vivo Biodistribution Assay with [^18^F]flumazenil

The results of the biodistribution study after the intravenous injection of 37 MBq/200 μL [^18^F]flumazenil showed reduced radioactivity in vivo. The %ID/g in the [^18^F]flumazenil/blood was 5.47 ± 0.56, 0.99 ± 0.42, 0.07 ± 0.07, 0.10 ± 0.11, 0.03 ± 0.04, and 0.01 ± 0.01 at 5, 10, 30, 60, 120, and 240 min, respectively (Figure 7).

At 5, 10, 30, 60, 120, and 240 min post-injection of [^18^F]flumazenil (37 MBq/200 μL), a rapid accumulation in the brain regions was observed at 5 min. The percentage of ID/organs of the brain was 1.92 ± 0.32, 0.39 ± 0.15, 0.03 ± 0.01, 0.03 ± 0.01, 0.001 ± 0.00, and 0.0004 ± 0.00 at 5, 10, 30, 60, 120, and 240 min, respectively. In the high density GABA_A_ regions of the brain, such as the cortex, hippocampus, and thalamus, the SBRs (with the pons/medulla as a reference region) were 1.96 ± 0.49, 1.55 ± 0.37, 1.37 ± 0.37 (at 5 min), 2.65 ± 0.85, 2.06 ± 0.41, 1.70 ± 0.40 (at 10 min), 5.32 ± 0.82, 3.24 ± 0.52, 1.37 ± 0.36 (at 30 min), and 2.22 ± 0.42, 1.78 ± 0.32, 1.57 ± 0.31 (at 60 min), as shown in Table 3. At 30 min post-injection, the highest SBR was reached, as compared to other internal times. The density pattern of GABA_A_ visualized in the PET experiments was consistent with the results of the biodistribution assay.

## 3. Discussion

Flumazenil is mainly metabolized and degraded by liver enzymes to reduce toxicity, and it is then excreted by the kidneys [5]. In the human liver, microsomal carboxylesterase isozymes play an important role in the detoxification and metabolism of flumazenil, and biotransforming it into two metabolites: flumazenil acid and N-demethylated flumazenil [14]. The biotransformation of xenobiotics can also be conducted using whole-liver homogenates to predict the types of reactions that may occur in vivo [15,16]. This in vitro drug metabolism model can adequately characterize drug metabolites and elucidate their pathways at concentrations similar to those in in vivo models. Gas chromatography and high-performance liquid chromatography (HPLC)-mass spectrometry-based methods have been used to determine flumazenil concentrations in plasma and urine, as well as its biodistribution in rodents and humans [10,11,12,17].

The metabolic path and rate of a drug are closely related to the distribution of the drug in the body, the rate of elimination, and the period of efficacy [2,3,5]. The study of drug metabolites is of great significance to ensure the proper prescription of drugs [4]. The mixed reaction of a test drug solution with animal tissue slices, homogeneous solutions, separation and extraction components, and liver cells can be used in metabolic reaction research [2,3,5]. Tandem mass spectrometry was used to identify flumazenil derivatives in liver homogenates and hepatic microsomal biosystems [9,15]. After the metabolic reaction, the solution was separated by HPLC-tandem MS analysis and analyzed by tandem MS to identify the metabolites [18]. The results of the tandem mass spectral analyses of the flumazenil fragments were consistent with the results reported by Amini and Leveque [11,19,20], and the fragment structures were also in line with those reported by Leveque [20]. However, the metastable fragmented ion structures of *m/z* 229, 217, and 189 corresponded to 5-ring delocalized imidazolium, 4-ring delocalized hydrazolium, and the loss of N2 fragments, respectively. However, this fragmentation pathway has not been reported previously.

During metabolism, a drug molecule is gradually transformed into more polar substances by metabolic enzymes; therefore, the R_t_ chromatography times of the metabolites are faster than those of the original drug molecules [2,3,5]. Its substructures are similar to those of the core fragment ions of flumazenil. The status of the metabolites M2 and M3 in the rat liver microsomal biosystem was determined (Figure 4). The parent ion of M2 had an *m/z* of 276, with fragmentation ions of 258 (Δ*m*/*z* = −18, –H_2_O) and 217, both of which were identical to the core fragmented ions of flumazenil. Mass spectra demonstrated that M2 has a molecular weight of 275, as well as the same substructure as that of flumazenil. M2 was, therefore, suggested to be the carboxylic acid form of flumazenil, or the hydrolyzed product of the ethyl ester. For the metabolite M3 (Figure 4B), the *m*/*z* = 320.4 of the molecular ion M3, +16 was higher than that of the parent molecule, demonstrating hydroxylation (+O) and the *m*/*z* of the fragmented ions (276 and 258), similar to the results for flumazenil, which extracted C_2_H_4_O (mw:44) and HOC_2_H_4_OH (mw:62), respectively, from M3. These mass data indicate that M3 likely results from the hydroxylation of the ethyl ester group of flumazenil to the hydroxyethyl ester. The hydroxylation of the methyl group is a type of phase I oxidation of hepatic enzymes. In flumazenil metabolism, biotransformation might occur in the N-methyl group of M1 (in rat liver homogenate solution) and M3 (in microsomal hepatic metabolic systems) to undergo demethylation (-CH3), resulting in the formation of N-methanol products with molecular weights of 289 and 319, respectively. This is the difference between the tandem mass spectra of the parent drugs.

In this study, fresh rat livers were homogenized and mixed with the flumazenil solution to conduct metabolic reactions, analyze its metabolites, and compare the differences in metabolic reactions between the liver-homogenized liquid and the liver microparticles. The flumazenil metabolism pathways identified in rat and human liver microsomes were similar to those identified by Amini in monkey microsomes [11], but differed from those identified in a rat liver homogenate. Our findings contrast with those of a previous metabolite analysis of carboxylesterase in a patient’s liver sample [14]. Here, the metabolite [N-demethylated Flumazenil] obtained from patient liver cells (P450 microsomal liver carboxylesterases) differed from the M1 obtained in mouse liver cells (the M1 compound was the same as that of human esterase + methanol). M2 is the preliminary metabolite (flumazenil acid) of esterase [14]; however, M3 was only obtained from mouse liver microsomes. Different species may present some differences in liver metabolic pathways; however, all liver microsomal isozymes metabolize lipophilic compounds into more hydrophilic carboxylic acids or alcohols [14]. The metabolic pathways of flumazenil in the three hepatic enzyme systems are summarized in Figure 5. Since the homogenized rat liver solution retains all liver enzyme functions, the influence of the liver matrix on the liver matrix disturbance of trace metabolite analysis may also be significant [21]. The metabolic rates and products of flumazenil metabolism in rat liver microsomes and rat liver homogenates were compared to determine the metabolic mechanism of flumazenil metabolism by liver enzymes [2,3].

Anxiety is a common emotional disorder that involves worrying about future events, and it is characterized by excessive and pointless worries, motor tension, and fatigue [13,22,23]. These are likely to be intensified due to nerve symptoms caused by the generated signals [24,25]. Accordingly, abnormalities in the gamma-aminobutyric acid (GABA) system in the brain correlate with the onset of epilepsy, anxiety, and other psychiatric disorders [13,26,27,28]. The GABA receptor, which is abundant in the cortex, is highly sensitive and prone to accidental damage. Several binding sites contain GABA receptors, including a benzodiazepine-binding site. Upon receiving a signal from GABA, a chloride channel is opened that allows the entry of Cl- into the nerve cells, thereby reducing the intracellular potential of GABA_A_ (type A) receptors [3,29,30].

The synthesis of [^18^F]flumazenil was conducted via a fast and convenient non-carrier nucleophilic substitution combined with the ^18^F- ion, with nitro-flumazenil as the precursor, using an automatic synthetic technique and solid extraction purification to replace the semi-preparative HPLC. Regarding in vitro serum stability, [^18^F]flumazenil showed a stable radiopurity when stored at 30% ethanol, with RCP (%) values of up to >85% at up to 1 h of incubation under regular conditions (Table 2). This compound stability is suitable for a radiopharmaceutical. Moreover, the high specificity and sensitivity of the target region could reduce the accumulated nonessential radiation dose (Figure 7). Regarding the metabolites, using flumazenil incubated in rat liver homogenate for 0–250 min resulted in a relatively low amount of M1 formation (Figure 3). After incubation with rat microsomes, the M2 metabolite formed was consistent with the decomposed parent flumazenil over time. However, in a previous study, isoflurane exhibited a high persistence of [^18^F]flumazenil uptake (up to 20%ID/g) in the brains of mice scanned under isoflurane anesthesia in static PET scans from awake animals, and this persistence was two-fold higher in the hippocampus of isoflurane-treated mice than in awake mice [31]. However, the use of isoflurane in small-animal PET imaging remains a recognized method. Regarding to the ex vivo biodistribution of [^18^F]flumazenil, the brain regions showed a rapidly declining trend from 5 to 10 min post-injection (Table 3). At different distribution times, the SBR ratio in the GABA_A_-abundant areas (such as the cortex, amygdala, and hippocampus), reached the highest SBR ratio in the cortex (amygdala) and hippocampus regions at a post-injected time of 30 min, relative to those of the pons/medulla (SBRs value 5.32 ± 0.82 and 3.24 ± 0.52, respectively) (Table 3). Because any metabolite is hydrophilic and it is difficult to penetrate the blood–brain barrier, based on the flumazenil kinetics achieved by biotransformation, we propose a post-injection window of 30–40 min for PET imaging, as the parent flumazenil will be more abundant at this time than any other metabolite. By utilizing [^18^F]flumazenil NanoPET/CT imaging, we observed both a global accumulation of GABA_A_ receptors, as well as a flumazenil metabolite compound, in rat brain regions [19,20]. No influence of [^18^F]flumazenil on the GABA_A_ receptors was observed (Figure 6). Regarding the biodistribution of [^18^F]flumazenil, the blood and brain regions showed a rapidly declining trend from 5 to 10 min post-injection (Figure 7).

This study had some limitations: 1. We used a traditional method for identifying flumazenil metabolism. However, we hope to directly analyze [^18^F]flumazenil in humans or primates, since different species may have different liver metabolism pathways. 2. The radiochemical purity under regular conditions after only 10 min was >90%, but the radiopurity needed to extend to less than 60 min for clinical use. However, we are still working toward improvements. The advantages of this study are: (1) In flumazenil metabolism, M1–3 biotransformation may occur in its nitrogen-methyl group for demethylation. (2) After administration of [^18^F]flumazenil for 30 min, we accumulated static PET images at 30–40 min, providing clear brain images, although the window for capturing images rapidly declined. (3) We demonstrated that the specific binding ratios of the GABA_A_ receptors were profoundly influenced in specific brain areas, including the amygdala, prefrontal cortex, cortex, and hippocampus, suggesting that these brain regions may be involved in the processing and storage of conditioned fear memories [13,19,32].

An increase in the binding capacity of the GABA_A_ benzodiazepine receptor site-specific radioligand (18F-flumazenil), using molecular imaging tracers to visualize the distribution of GABA_A_ receptors in the brain, was employed to aid in the diagnosis of psychiatric/CNS disorders (such as epilepsy, anxiety, and insomnia) [6,13]. It can be traced by radioactive flumazenil derivatives that are more sensitive and accurate than 2-deoxy-2[^18^F]fluoro-D-glucose ([^18^F]FDG) imaging that is often used for epileptic foci localization [5,7]. In the present study, the integration of flumazenil metabolites into rat liver homogenates and microsomes was analyzed using liquid chromatography-tandem mass spectrometry. A complete understanding of the in vivo metabolism of flumazenil will accelerate drug development for inspection and registration purposes. We characteristically showed completion of the biology of the final product with [^18^F]flumazenil by in vitro, ex vivo and in vivo imaging. [^18^F]flumazenil is a good ligand and can serve as a PET agent for the determination of the GABA_A_/BZR complex for multiplex neurological syndromes in future clinical stages.

## 4. Materials and Methods

### 4.1. Materials and Reagents

Analytical-grade laboratory chemicals for LC/MS were used as received, without further purification. Methanol, acetonitrile (ACN) (both HPLC and MS grade), ammonium acetate, ethyl acetate (EA), phosphate buffer pellets, sucrose, and bovine serum albumin (BSA) were purchased from Merck (Darmstadt, Germany). Flumazenil was purchased from FutureChem (Seoul, Republic of Korea). Deionized water was prepared using a Smart DQ3 reverse osmosis reagent water system (Merck Millipore, Billerica, MA, USA) with a 0.22 μm polyvinylidene fluoride (PVDF) filter and a UV light source to produce ultra-pure water (total organic carbon < 5 ppb, resistivity ≥ 18.2 MΩ-cm). A ZORBAX Eclipse XDB-C18 reversed phase HPLC column (100 × 4.6 mm, particle size:5 μm) from Agilent Technologies (Palo Alto, CA, USA) was used to study flumazenil and its metabolites.

### 4.2. Animals

All experiments were performed on Sprague Dawley (SD) rats (3–4 weeks old) purchased from a licensed breeder (BioLASCO Taiwan Co., Taipei, Taiwan). All the animal procedures and experimental protocols were approved by the Ethical Animal Use Committee of the Institute of Nuclear Energy Research (INER), Atomic Energy Council of Taiwan. This study complied with Taiwan’s laws on the care and use of laboratory animals (LAC-2019-0253). The rats were maintained at 21 ± 2 °C with 50 ± 20% relative humidity under an alternative 12 h light and 12 h dark cycle. The rats were identified as normal using NanoPET/CT imaging, and the serum stability and biodistribution of their brains were evaluated. Fresh rat livers were obtained from healthy male SD rats and frozen at −70 °C. Rat liver microsomes and co-enzymes (NADPH A and B) were purchased from BD Biosciences (Bedford, MA, USA) and stored at −70 °C.

### 4.3. Apparatus and Equipment

#### 4.3.1. High Performance Liquid Chromatography Instrumentation

The quantification of flumazenil metabolism by hepatic enzymes was performed using HPLC (Agilent 1100 series, Palo Alto, CA, USA), which is comprised of an online degasser, a binary pump, an auto sampler, a thermostat column oven (maintained at 25 °C), and a diode array detector (DAD, 250 nm). Data were acquired and processed using the Agilent ChemStation software (ed. 10.02). Separation was achieved using a C18 column with a guard column and an ammonium acetate buffer–acetonitrile gradient elution. The mobile phase consisted of eluents A (5 mM ammonium acetate aqueous buffer with 1% ACN, pH 7.0) and B (ACN only) at a flow rate of 0.6 mL min^−1^. These were mixed to separate the metabolites and flumazenil from the sample matrices, according to the gradient program shown in Table 1. The overall LC time for flumazenil biotransformation was 15 min. An analyte solution of 5 µL was injected using an auto-sampler thermostat at 4 °C.

#### 4.3.2. Liquid Chromatography-Mass Spectrometry/Mass Spectrometry Instrumentation

Mass spectrometric analysis of flumazenil in ACN was initially conducted on a 4000 QTrap LC-MS/MS system using API Analyst software 1.4.1 (AB Sciex, Concord, ON, Canada) to obtain the appropriate MS parameters and *m*/*z* for flumazenil fragments. Samples were introduced into the spectrometer, either by the HPLC system (Agilent 1100) or by a syringe pump, at a flow rate of 10 µL min^−1^ (Harvard Apparatus Inc., Holliston, MA, USA). Chromatographic separation was conducted as described in the HPLC conditions section. The analytes were ionized by a turbo spray ion source (electrospray ionization, [ESI]) in the positive-ion mode at 5000 V and 350 °C. Mass spectra were recorded within a range of 100–1000 amu with unit resolutions of Q1 and Q3. The other experimental parameters are listed in Table 1. The LC-tandem MS determination of flumazenil concentration was conducted in the multiple reaction monitoring positive-ion mode with transition ion pairs 304 → 258 and 304 → 276. Vaporized liquid nitrogen gas was used as the nebulizer, curtain, and collision gas in all 4000 QTrap LC–MS/MS studies.

### 4.4. Procedures for Metabolism Study

#### 4.4.1. Biotransformation and Pretreatment for High Performance Liquid Chromatography of Flumazenil in Rat Liver Homogenate

Normal rat livers were obtained from healthy male SD rats weighing 400–450 g. The samples were homogenized using the following procedure. The rats were euthanized by CO_2_ inhalation for 5 min in a closed box. After confirmation of euthanization, the entire liver (approximately 14 g) was removed, washed with ice-cold normal saline solution, and immediately minced in a 0.25 M sucrose solution (30 mL) in an ice bath. The rat liver tissue solution (40 mL) was homogenized in a motor-driven homogenizer, aliquoted (1.5 mL) into sample tubes, and stored at −70 °C. The samples were used for biotransformation studies within 30 d and thawed in an ice bath before use. For the biotransformation, the liver homogenate (1 mL) was placed in a 15 mL test tube and mixed well with a 1:1 (*v*/*v*) solution of 0.25% BSA in 0.01 M phosphate-buffered saline solution (1 mL, pH 7.4). The flumazenil solution intended for study (10 μL volume; 1000 μg mL^−1^ in dimethyl sulfoxide [DMSO]) was added to the tube, which was swirled completely and placed in a preheated water bath (37 °C, 70 rpm) to initiate the enzymatic reaction. After specified incubation periods (0, 10, 20, 30, 45, 60, 90, 120, and 240 min), ACN (2 mL) was added to each tube to stop the reaction and precipitate proteins, which were then centrifuged (10,000 rpm, 5 min, 4 °C) to separate the supernatants. Liquid–liquid extraction with EA (400 μL) was performed in triplicate to concentrate the flumazenil in the supernatant and desalt the reaction mixture. The EA fraction (1.2 mL) was collected in a microtube and purged with N_2_ (thermostat set at 30 °C) to vaporize the EA until the volume was reduced to less than 50 μL. The samples were again dissolved in ACN to generate a final volume of 1.0 mL and filtered through a 0.22 µm PVDF disk filter for LC-DAD/tandem MS analysis. The liver homogenate (2 mL) was mixed well with a buffer solution (2 mL) and DMSO (10 µL) to generate a blank background solution.

#### 4.4.2. Flumazenil Metabolization in Rat Liver Microsomes

Incubation solutions were prepared according to the manufacturer’s instructions. First, 2 μL of a 5-mM flumazenil solution in DMSO was added to a reaction mixture buffered with 0.5 M potassium phosphate pH 7.4 (200 μL) containing co-enzyme NADPH solution A (50 μL) and B (10 μL) and 25 μL of liver microsome protein solution (0.5 mg). The final volume of the incubated solution was adjusted to 1.0 mL with deionized water (713 μL). The reaction vials were capped, mixed well, and placed in a pre-heated water bath (37 °C, 70 rpm) to initiate enzymatic reactions. After incubating, the reactions for specified durations (0, 10, 20, 30, 45, 60, 90, and 120 min), 100 μL of each incubated solution was mixed well with 100 μL of ACN in microcentrifuge tubes to terminate the enzymatic reactions. This was followed by centrifugation to precipitate proteins (10,000× *g*, 4 min, 4°C). The supernatants were filtered through a 0.22 µm PVDF disk filter and analyzed using LC-DAD and tandem MS. The liver microsome solution (25 μL) was also mixed with DMSO (2 μL without flumazenil), buffer, and the co-enzyme solution to generate a blank solution.

### 4.5. Synthesis of [^18^F]flumazenil

[^18^F]flumazenil was prepared by a nucleophilic substitution reaction using an automated Tracerlab FX-FN cartridge synthesizer (General Elextric Company, Necco St, Boston, MA, USA). [^18^F]flumazenil was prepared according to a modified version of a previously described method [2,33]

### 4.6. In Vitro Serum Stability

Serum preparation: healthy SD rats were deeply anesthetized with isoflurane gas (3% isoflurane in 50% oxygen, 1 mL/min). One milliliter of whole blood from healthy SD rats was withdrawn through the tail vein at room temperature and centrifuged at 3000 rpm for 10 min to obtain the prepared serum for stability testing.

Stability in serum: the stability was studied by incubation of 3.7 MBq of [^18^F]flumazenil in 200 uL of prepared serum at 37 °C. At different time points (0, 5, 10, 30, 60, 120, and 240 min), each sample was treated with 1 mL of acetonitrile to precipitate serum proteins. After mixing and centrifugation for 2 min at 5000 rpm, the radiochemical purity of [^18^F]flumazenil was determined using radio-TLC. Radio-TLC was performed on a silica gel plate TLC-SG (1.5 × 6 cm), using ethyl acetate: ethanol (80/20) as the developing agent.

### 4.7. In Vivo NanoPET/CT Neuroimaging Studies

The rats were injected with approximately 37 MBq/200 μL of [^18^F]flumazenil via the lateral tail vein. After being distributed for 30 min, isoflurane gas (3%) was used to anesthetize the animals, which were then placed inside a NanoPET/CT PLUS system (Bioscan Europe, Ltd., Paris, France) for statistical analysis (post-injection 30–40 min). InVivoScope software program (Bioscan Inc.) was used to capture the NanoPET images. Pmod 3.3 software was used to merge the NanoPET/CT images and set them in the brain template. A region of interest (ROI) (cerebellum, medulla, pons, midbrain, striatum, frontal cortex, cortex, amygdala, and hippocampus) was defined in each brain region on the NanoPET images using coronal sections. The average radioactivity concentration within each brain section was calculated by averaging the pixel values within the multiple ROIs. When analyzing the NanoPET data, the medulla was used as the reference region to determine the specific binding ratios for all target regions.

### 4.8. Ex Vivo Bio-Distribution Assay

While the rats were anesthetized with isoflurane, [^18^F]flumazenil (37 MBq/200 μL) was injected directly into the tail vein. The rats (*n* = 3 at each time point) were sacrificed by the administration of carbon dioxide at 5, 10, 30, 60, 120, and 240 min post-injection. Different brain regions and blood samples were removed and weighed, and radioactivity was evaluated using an automatic gamma counter (WIZARD2 2480, PerkinElmer Instruments Inc., Waltham, Massachusetts, United States). The percentage dose per organ was calculated by comparing tissue counts with suitably diluted aliquots of the injected material. Different brain regions corresponding to the cerebellum, pons/medulla, striatum, cortex/amygdala, hippocampus, thalamus, and hypothalamus were excised. The % dose/g and specific binding ratio (SBR) of the samples were calculated by comparing sample counts with the diluted initial dose.

## 5. Conclusions

In conclusion, the flumazenil metabolic pathways were identified in rat liver microsomes and homogenates by HPLC combined with ESI-QqQ MS, revealing the mechanism of flumazenil biotransformation. A new metabolite compound with a methyl replacement at the ethyl ester is thought to be a novel flumazenil metabolite. Through an automatic synthesis process, the successful radiofluorination of [^18^F]flumazenil and its utilization in PET showed evidence of an animal model expressing the GABA_A_/BZR receptor. Although the distribution of [^18^F]flumazenil in metabolism analysis showed about 50% biotransformed metabolites after 30 min, ex vivo biodistribution and in vivo imaging studies still showed significant differences in the target regions. The remaining complete compound structure can continuously accumulate in the target regions of the brain in order to facilitate molecular nuclear medicine imaging. A complete analysis of flumazenil biotransformation by hepatocytes may provide evidence of the efficacy of [^18^F]flumazenil for use in patients with anxiety at the clinical stage.

## Figures and Tables

**Figure 1 pharmaceuticals-16-00764-f001:**
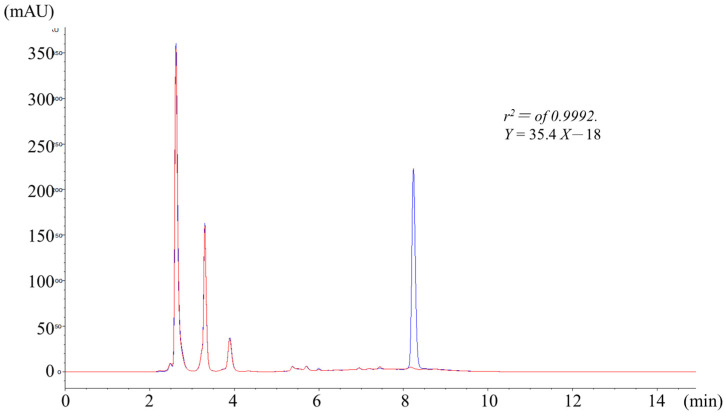
Typical chromatogram of flumazenil exhibiting a spike in the rat liver homogenate. HPLC analysis using ZORBAX Eclipse XDB-C18 column separated flumazenil from the liver microsome and the liver homogenate solutions (DAD at λ of 250 nm). The dynamic range of flumazenil in the hepatic matrices was 1–100 ppm, and the limits of detection (LOD) and quantification (LOQ) for flumazenil were 0.1 and 0.4 ppm, respectively.

**Figure 2 pharmaceuticals-16-00764-f002:**
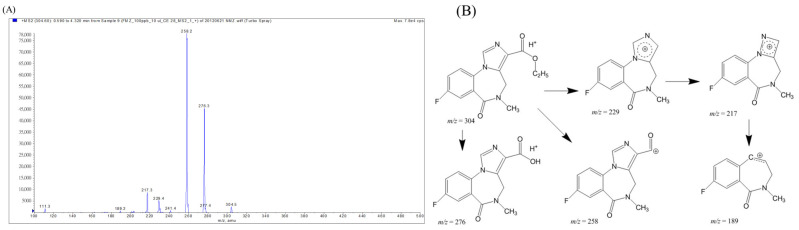
Mass spectrum of fragmentation ions from *m*/*z* = 304 (dissolved in methanol with 0.1% formic acid, 10 µL min^−1^) injected into the instrument for the Q1 and MS/MS scans. (**A**) product-ion scans using different collision-activated dissociation conditions to optimize the de-clustering potential fragmentation scheme of flumazenil based on the tandem mass spectrum. (**B**) The fragment structures of flumazenil with metabolite identification; the most abundant product ions were 258 and 276, from the parent molecular ion 304, and were also observed at *m/z* values of 229, 217, and 189, respectively.

**Figure 3 pharmaceuticals-16-00764-f003:**
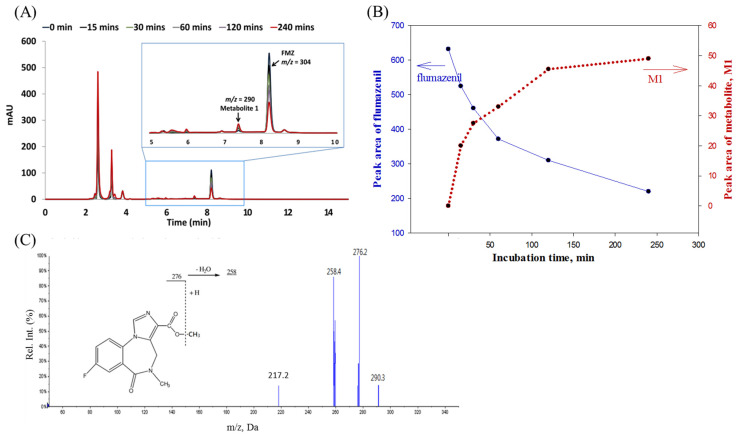
(**A**) After pre-treatment with flumazenil concentrate for 0 to 240 min, the derivatives from the hepatic matrices were analyzed by HPLC. (**B**) Overlapped chromatograms showing the analysis of flumazenil incubation in rat liver homogenate for various durations. (**C**) The flumazenil biotransformation in the rat liver homogenate was evaluated based on trends in the peaks of flumazenil, as shown on the chromatograms. In the first 30 min, biotransformation subsequently slowed down, reaching approximately 50% at 60 min and 65% at 4 h.

**Figure 4 pharmaceuticals-16-00764-f004:**
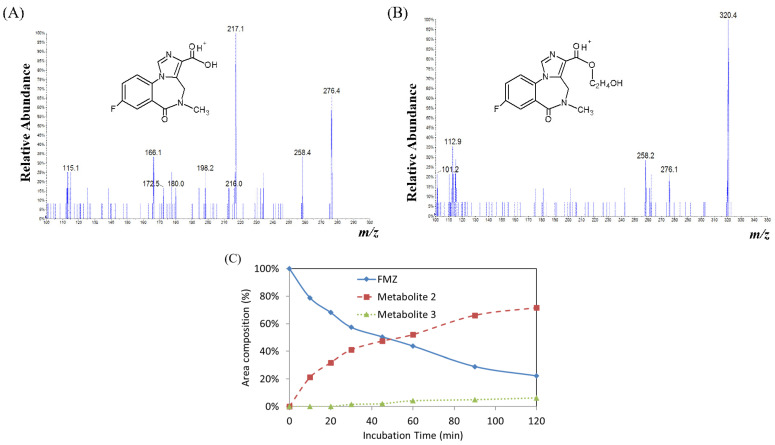
Flumazenil metabolism study in human liver microsomes. (**A**,**B**) Tandem mass spectra of M2 and M3, *m*/*z* = 276 and 320, and proposed metabolites of flumazenil in rat microsomes, respectively. (**C**) Time course of biotransformation of flumazenil in rat liver microsomes for various durations.

**Figure 5 pharmaceuticals-16-00764-f005:**
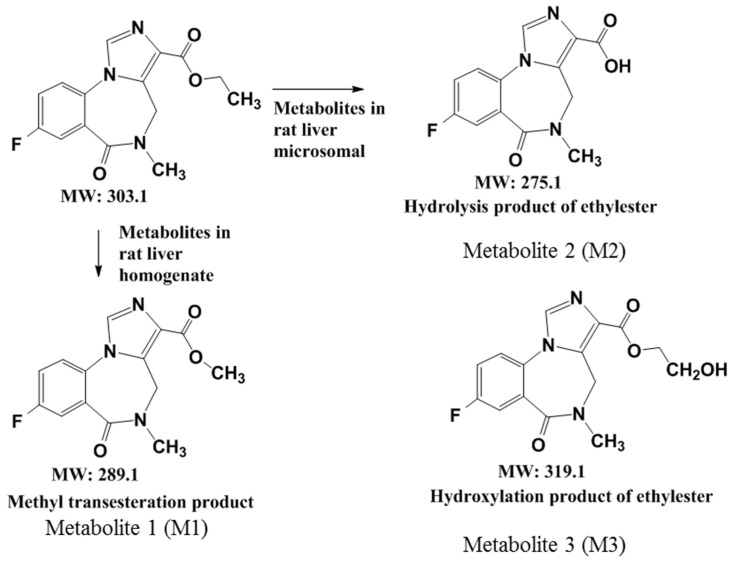
A summary of the metabolic pathways of flumazenil in the rat liver homogenate and the microsomal enzymatic system.

**Figure 6 pharmaceuticals-16-00764-f006:**
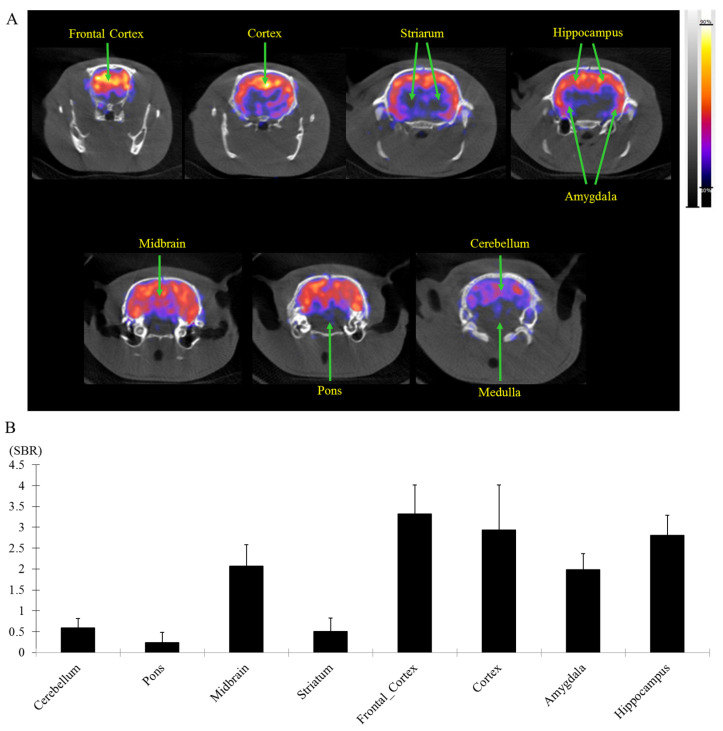
The brain regional distribution of GABA_A_ receptors as imaged by [^18^F]flumazenil NanoPET/CT in normal SD rats. The static images were acquired using NanoPET/CT at 30–40 min. The rats displayed a markedly higher uptake of [^18^F]flumazenil in the prefrontal cortex, cortex, amygdala and the hippocampus; the medulla was delineated as the ROI, and was used as a reference region. (**A**). In vivo NanoPET/CT image of the coronal views of [^18^F]flumazenil at 30–40 min. (**B**). The specific uptake ratio (SBR) in different brain regions at 30–40 min post-injection showed a significant cerebral uptake of [^18^F]flumazenil.

**Figure 7 pharmaceuticals-16-00764-f007:**
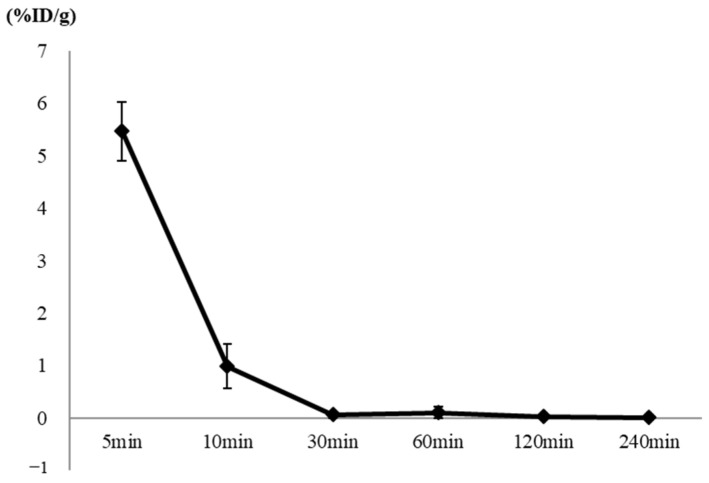
The %ID/g in the [^18^F]flumazenil blood distribution analysis was determined using an automatic gamma counter. The percentage dose per gram was calculated by a comparison of the blood counts with suitably diluted aliquots of the injected material. The blood %ID/g of [^18^F]flumazenil was 5.47 ± 0.56, 0.99 ± 0.42, 0.07± 0.07, 0.10 ± 0.11, 0.03 ± 0.04, and 0.01 ± 0.01 at 5, 10, 30, 60, 120, and 240 min, respectively.

**Table 1 pharmaceuticals-16-00764-t001:** High performance liquid chromatography (HPLC) and tandem mass spectrometry instrumental parameters for flumazenil biotransformation.

Parameters	
HPLC	
Stationary phase	ZORBAX Eclipse XDB-C18 4.6 ID × 100 mm, 5 μm thermostat at 25 °C
Mobile phase	
Flow rate	0.6 mL min^−1^
Composition	A: ammonium acetate 5 mM aqueous pH 7.0 with 1% CH_3_CN
	B: only CH_3_CN
Gradient program	0–3.5 min, 10% → 40% B
	3.5–10.0 min, 40% B isocratic
	10.0–10.1 min, 40% → 10% B
	10.1–15.0 min, 10% B isocratic
Detector	DAD at 250 nm
Turnaround time	17 min
Mass Spectrometry	
Source temperature (°C)	350
Polarity	Positive
Resolution, Q1 and Q3	Unit
Nebulizer gas, NEB (psi)	40
Curtain gas, CUR (psi)	10
Turbo gas	15
Collision gas, CAD (psi)	Medium
Ion spray voltage, IS (V)	5000
Ion energy 1, IE1 (V)	0.4
Ion energy 3, IE3 (V)	0.3
Declustering potential, DP (V)	70
Entrance potential, EP (V)	10
Detector parameter	Positive
-Channel electron multiplier, CEM (V)	1950
Multiple reaction monitoring (MRM) transition pair	304 > 258 and 304 > 276

**Table 2 pharmaceuticals-16-00764-t002:** In vitro serum stability of [^18^F]flumazenil. Prepared serum for incubation of 3.7 MBq of [^18^F]flumazenil at 37 °C at different time points (0, 5, 10, 30, 60, 120, and 240 min), treated with 1 mL of acetonitrile for precipitating the serum proteins. The radio-chemical purity of [^18^F]flumazenil was determined using radio-TLC methods. Radio-TLC was performed on a silica gel plate TLC-SG (1.5 × 6 cm), using ethyl acetate: ethanol (80/20) as the developing agent.

Time (Min)	Radiochemical Purity (RCP, %)
0	95.76
5	93.28
10	91.07
30	87.46
60	85.68
120	79.95
240	72.13

**Table 3 pharmaceuticals-16-00764-t003:** Ex-vivo biodistribution of [^18^F]flumazenil at 5, 10, 30, 60, 120, and 240 min after injection in SD rats. The percentage dose per gram (%ID/g) was calculated by comparing the sample counts with the count of the diluted initial dose. In the calculated SBR, different brain regions are compared to the pons/medulla.

%ID/g	5 min	10 min	30 min	60 min	120 min	240 min	
Cerebellum	2.22 ± 0.12	0.74 ± 0.04	0.07 ± 0.00	0.02 ± 0.00	0.01 ± 0.00	0.00 ± 0.00	
Pons/Medulla	1.80 ± 0.19	0.60 ± 0.06	0.06 ± 0.01	0.02 ± 0.00	0.01 ± 0.00	0.00 ± 0.00	
Striatum	3.27 ± 0.32	1.09 ± 0.11	0.11 ±0.01	0.04 ± 0.00	0.02 ± 0.00	0.01 ± 0.00	
Cortex (Amyglada)	5.28 ± 0.60	2.16 ± 0.39	0.38 ± 0.02	0.06 ± 0.01	0.03 ± 0.00	0.01 ± 0.00	
Hippocampus	4.57 ± 0.56	1.82 ± 0.19	0.25 ± 0.01	0.05 ± 0.00	0.03 ± 0.00	0.01 ± 0.00	
Thalamus	4.22 ± 0.41	1.61 ± 0.14	0.14 ± 0.00	0.05 ± 0.00	0.02 ± 0.00	0.01 ± 0.00	
Hypothalamus	3.71 ± 0.18	1.34 ± 0.06	0.12 ± 0.00	0.04 ± 0.00	0.02 ± 0.00	0.01 ± 0.00	
**SBR**	**5 min**	**10 min**	**30 min**	**60 min**	**120 min**	**240 min**	
Cerebellum	0.25 ± 0.18	0.36 ± 0.26	0.16 ± 0.15	0.36 ± 0.14	0.32 ± 0.07	0.71 ± 0.32	
Striatum	0.84 ± 0.37	0.92 ± 0.34	0.91 ±0.29	1.00 ± 0.36	0.94 ± 0.24	0.98 ± 0.33	
Cortex (Amyglada)	1.96 ± 0.49	2.65 ± 0.85	5.32 ± 0.82	2.22 ± 0.42	2.14 ± 0.43	1.71 ± 0.51	
Hippocampus	1.55 ± 0.37	2.06 ± 0.41	3.24 ± 0.52	1.78 ± 0.32	1.71 ± 0.41	1.73 ± 0.87	
Thalamus	1.37 ± 0.37	1.70 ± 0.40	1.37 ± 0.36	1.57 ± 0.31	1.51 ± 0.30	1.53 ± 0.58	
Hypothalamus	1.08 ± 0.26	1.25 ± 0.28	1.15 ± 0.14	1.26 ± 0.21	1.20 ± 0.19	1.02 ± 0.42	

## Data Availability

Data is contained within the article.

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
