# Peer review of "Identification of the Hepatic Metabolites of Flumazenil and their Kinetic Application in Neuroimaging"

_pharmaceuticals, 2023, doi:10.3390/ph16050764_

Round 1

Reviewer 1 Report

Clinical studies have demonstrated that the GABAA receptor complex plays a central role in the modulation of anxiety. The present study presents [18F]flumazenil as a good radiopharmaceutical for PET and analyzes its three metabolites with the aim of speeding up the registration process. The analysis of liver metabolites is studied indirectly on the ligand and not on the radiopharmaceutical using rat liver homogenate and liver microsomes and subsequently separating and characterizing the metabolites. Then the authors try to correlate the radiopharmaceutical distribution kinetics of the presented imaging with the formation of the indirectly identified metabolites. It would also have been interesting to see the radioactive metabolites analyzed in an analytical HPLC equipped with a radioactive detector.

The work is interesting and fits into a fairly recent line of research. I ask the AUTHORS to make the title more specific (better the "running title") and to make the objective of the study clearer to the readers in the abstract, in the discussion and also for coherence of the discussion. The biodistribution data of the radiopharmaceutical presented have already been the subject of another publication, here the weight on the kinetics of the discovery of the metabilites performed must be more highlighted.

Author Response

Response to Reviewer 1 Comments

Point 1: I am not qualified to assess the quality of English in this paper

Response 1: Thanks for reviewer’s suggestion. This article had editing certificate from Editage company. If the quality of English still not enough, I could editing the English language again.

Point 2: I ask the AUTHORS to make the title more specific (better the "running title") and to make the objective of the study clearer to the readers in the abstract, in the discussion and also for coherence of the discussion. The biodistribution data of the radiopharmaceutical presented have already been the subject of another publication, here the weight on the kinetics of the discovery of the metabolites performed must be more highlighted.

Response 2: Thanks for reviewer’s suggestion. We had modified the title to ‘Identification of the hepatic metabolites of flumazenil and utilize kinetics application in neuroimaging’, and running title to ‘Identification flumazenil metabolites and utilize kinetics in neuroimaging’ wish to fit the content.

In the whole article, we weight some kinetic discovery by 'track changes' function wish could improve the past shortcomings.

Reviewer 2 Report

K. W. Chang et.al. studied the metabolites of flumazenil in hepatic system based on HPLC-MS analysis. This interesting study will help for radiopharmacists to figure out the actual structure of the compound which binds to the target region.

some comments on the manuscript as below.

- page 7, line 23, Figure 5, metabolic pathway of Flumazenil in rat liver, the aryl fluoride is missing in structure of (M3).

- Author has to comments on the possible way of formation metabolite M3  in rat liver.

-page 13, line 48, 4.5 Synthesis of [18F]flumazenil, details procedure [18F] reaction with scale of nitro-precursor, RCY & HPLC data of synthesized [18F]flumazenil need to mentioned in this section.

Author Response

Response to Reviewer 2 Comments

Point 1: - page 7, line 23, Figure 5, metabolic pathway of Flumazenil in rat liver, the aryl fluoride is missing in structure of (M3).

Response 1: Thanks for reviewer’s suggestion. We fix the mistake.

Point 2: Author has to comments on the possible way of formation metabolite M3 in rat liver.

Response 2: Thanks for reviewer’s suggestion. We have added some descriptions in discussion.

Point 3: page 13, line 48, 4.5 Synthesis of [18F]flumazenil, details procedure [18F] reaction with scale of nitro-precursor, RCY & HPLC data of synthesized [18F]flumazenil need to mentioned in this section.

Response 3: Thanks for reviewer’s suggestion. We have added some descriptions in results 2.5.

Reviewer 3 Report

Dr Chen and colleagues describe the whole body distribution and metabolism of [18F]flumazenil in rat and in vitro models. Flumazenil (FMZ) is a drug that is a commonly used benziodiazepine antagonist. FMZ-PET has been studied as a potential clinical tool for the localization of epilepsia and other indications. Therefore, the topic is of interest. However analyses performed in this manuscript are not novel, the methodology need several updates and the manuscript is badly structured. I have the following suggestions to improve:

1)      The introduction should be fully restructured. The first paragraph may be either fully removed or removed to the discussion when discussing potential applications of this imaging biomarker.

2)      Paragraph 2-4 do not lead to the aim of the study, please clarify what is known and why this study was performed in a clear aim.

3)      The in vitro model resembles an earlier in vitro metabolism study and shows the same metabolic profile: https://pubmed.ncbi.nlm.nih.gov/10942971/. Can the authors discuss this manuscript in the introduction and describe what is new in the current in vitro study?

4)      It remains unclear why only Sprague Dawley Rats were chosen and no diseased animal model was considered (especially as currently multiple diseases are introduced in the first paragraph). In the past, multiple rodent PET studies have been performed.

5)      In the PET sample analyses, total radioactivity was measured. Here, the authors did not account for the rapid metabolism of the  [18F]flumazenil that they observed. This is a large caveat of the PET analyses. Metabolism should be included in the PK modeling of the data and the formation of metabolites  should be discussed.

6)      It remains unclear what the binding affinity is of the metabolites to  the GABA receptor. This is a large study limitation, specifically for the PET study of total radioactivity. Please discuss.

7)      The result section repeats some of the methodology paragraphs (eg page 5 row 161-164). Please restructure the result section.

8)      Radiolabeled flumazenil PET has been studied in rodents (eg https://ejnmmires.springeropen.com/articles/10.1186/s13550-016-0235-2, https://www.ncbi.nlm.nih.gov/pmc/articles/PMC3348032/ ) Please discuss these prior analyses and discuss similarities and differences. 

9)      The discussion has many repetitions. A thorough review English language and restructuring of the discussion is needed: It should include what are the most important findings, what is known and what is new.

It looks like some parts have been copied from other manuscripts as some paragraphs are highly off topic, but the English language is correctly used, whereas for the paragraphs containing results and limitations is hard to read due to the language barrier.

Author Response

Response to Reviewer 3 Comments

Point 1: The introduction should be fully restructured. The first paragraph may be either fully removed or removed to the discussion when discussing potential applications of this imaging biomarker.

Response 1: Thanks for reviewer’s suggestion. We had modify introduction by 'track changes' function wish could improve the past shortcomings.

Point 2: Paragraph 2-4 do not lead to the aim of the study, please clarify what is known and why this study was performed in a clear aim.

Response 2: Thanks for reviewer’s suggestion. In the final on introduction, we had described more clear aim about this article.

Point 3: The in vitro model resembles an earlier in vitro metabolism study and shows the same metabolic profile: https://pubmed.ncbi.nlm.nih.gov/10942971/. Can the authors discuss this manuscript in the introduction and describe what is new in the current in vitro study?

Response 3: Thanks to Reviewer for giving the reference. In this article explored to the metabolite analysis of carboxylesterase by a patient liver sample. The author based on liver sample obtained two compounds, which are different from what our founding. Among them, the metabolite obtained from patient liver cells (P450 microsomal liver carboxylesterases) is different from the M1 we obtained in mouse liver cells (and M1 compound was the same with the paper consistent with human esterase + methanol). In addition, the M2 and M3 metabolites we obtained in the mouse liver microsomal are also the part that the author did not carry out before, but M2 is the same as the preliminary metabolite (flumazenil acid) of esterase. We have added some descriptions in the discussion.

Point 4: It remains unclear why only Sprague Dawley Rats were chosen and no diseased animal model was considered (especially as currently multiple diseases are introduced in the first paragraph). In the past, multiple rodent PET studies have been performed.

Response 4: Thanks for reviewer’s suggestion. In this article, we went to show the metabolites of flumazenil, and also conduct what’s the future plan on this radiopharmaceutical (for PET image). So in this article we chose the normal rat for study, instead of diseased animal model would be more coincidence with normal metabolic conditions

Point 5: In the PET sample analyses, total radioactivity was measured. Here, the authors did not account for the rapid metabolism of the [18F]flumazenil that they observed. This is a large caveat of the PET analyses. Metabolism should be included in the PK modeling of the data and the formation of metabolites  should be discussed.

Response 5: Thanks for reviewer’s suggestion. We have added some descriptions in the discussion.

Point 6: It remains unclear what the binding affinity is of the metabolites to the GABA receptor. This is a large study limitation, specifically for the PET study of total radioactivity. Please discuss.

Response 6: Thanks for reviewer’s suggestion. We had modify discussion by 'track changes' function wish could improve the past shortcomings.

Point 7: The result section repeats some of the methodology paragraphs (eg page 5 row 161-164). Please restructure the result section.

Response 7: Thanks to the reviewer for the suggestion. We removed some redundant words in the results.

Point 8: Radiolabeled flumazenil PET has been studied in rodents (eg https://ejnmmires.springeropen.com/articles/10.1186/s13550-016-0235-2, https://www.ncbi.nlm.nih.gov/pmc/articles/PMC3348032/ ) Please discuss these prior analyses and discuss similarities and differences.

Response 8: Thanks to the reviewer for the suggestion. In reviewer supply paper, it discusses on different anesthetic agents will effect GABAA receptor binding efficiency on F-18-FMZ. This is different direction between my discussed in this article, but I still refer to this article to add what kind of impact when used isoflurane for anesthetic.

Point 9: The discussion has many repetitions. A thorough review English language and restructuring of the discussion is needed: It should include what are the most important findings, what is known and what is new.

Response 9: Thanks for reviewer’s suggestion. We had modify introduction by 'track changes' function wish could improve the past shortcomings.

Point 10: It looks like some parts have been copied from other manuscripts as some paragraphs are highly off topic, but the English language is correctly used, whereas for the paragraphs containing results and limitations is hard to read due to the language barrier.

Response 10: Thanks for reviewer’s suggestion. This article had editing certificate from Editage company. If the quality of English still not enough, I could editing the English language again.

Round 2

Reviewer 3 Report

The manuscript has improved. However, further clarification and more English editing needs to be performed. Below, I provide some suggestions for the authors, but there are more textual errors that need English editing. Therefore, please, continu to edit the English language.

My specific suggestions are:

In the abstract, please change: “ And a carrier-free nucleophilic fluorination with automatic synthesizer for [18F]flumazenil which applied to in vivo nano-positron emission tomography (NanoPET)/computed tomography (CT) imaging and ex vivo bio-distribution used to analyze in normal rats.” To something like “ A carrier-free nucleophilic fluorination with automatic synthesizer for [18F]flumazenil combined with nano-positron emission tomography (NanoPET)/computed tomography (CT) imaging was used to predict the bio-distribution in normal rats.

In the introduction, the aim (row 134, page 3) remains unclear and could be transformed from: “ In this article, .. syndromes in the clinical stage” to something like “ This study aims to clarify the bio-transformation of flumazenil using in vivo and in vitro hepatic systems and to quantify the GABAA/BZR complex distribution of [18F]flumazenil by Nano-positron emission tomography (NanoPET). When fully characterized, PET based GABAA/BZR images provide a potential biomarker for diagnosis and treatment of patients with multiple neurological syndromes.

Furthermore, the introduction remains misleading: Paragraph 1 does not relate to the topic discussed in the manuscript. I would advise to move the first paragraph to the discussion and start with an introduction of the current use of flumazenil and why there is a need to characterize its metabolism and whole body distribution in preclinical models.

Furthermore, some more work should be done in the discussion:

The added sentence row 337 page 11 “ In this model, all enzymes .. from the biotransformation system” is unclear. Please rephrase, eg to: “This in vitro drug metabolism model, can adequately characterize drug metabolites and elucidate their pathways in concentrations resembling the in vivo models.”

The sentence page 12, row 499 is somewhat unclear: “Compare with results reported by Britta’s work, .. hydrophilic carboxylic acids or alcohols [27]..”

This can be improved, eg to; Our findings differ from prior metabolite analysis of carboxylesterase by a patient liver sample [27]. Here, the metabolite [please name] obtained from patient liver cells (P450 microsomal liver carboxylesterases) differs from the M1 we obtained in mouse liver cells (and M1 compound was the same as consistent with human esterase + methanol). M2 is the same as the preliminary metabolite (flumazenil acid) of esterase [27], but M3 was only obtained in our mouse liver microsomes. Different species maybe present some differences in liver metabolism pathways, but all liver microsomal isozymes will metabolize lipophilic compounds to more hydrophilic carboxylic acids or alcohols [27].

Furthermore, please remove the sentence row 403: “And more hydrophilic metabolite will unfavorable penetrate blood brain barrier, disadvantageous for radiopharmaceutical accumulated in specific brain regions.”

Furthermore in the discussion, please do not repeat results by repeating results, eg listing metabolites and please refrain from referring to tables and figures in the Discussion. For example, please move the full paragraph of page 11 row 361 “ In a rat liver homogenate solution… mass spectra from the parent drug.”to the result section (to paragraph 2.3.1 and 2.3.2).

See above

Author Response

Thanks for reviewer’s suggestion. I had follow all suggestion to modify the article by 'track changes' function to improve the past shortcomings. Re-editing the article from Editage company about the English revision, but the original 'track changes' function were disappeared.
